# Shotgun Metagenomic Sequencing Analysis as a Diagnostic Strategy for Patients with Lower Respiratory Tract Infections

**DOI:** 10.3390/microorganisms13061338

**Published:** 2025-06-09

**Authors:** Ha-eun Cho, Min Jin Kim, Jongmun Choi, Yong-Hak Sohn, Jae Joon Lee, Kyung Sun Park, Sun Young Cho, Ki-Ho Park, Young Jin Kim

**Affiliations:** 1Department of Laboratory Medicine, Graduate School, Kyung Hee University, Kyung Hee University Hospital, Seoul 02447, Republic of Korea; ha-eun0111@naver.com; 2Department of Laboratory Medicine, Seegene Medical Foundation, Seoul 04805, Republic of Korea; lithium2864@mf.seegene.com (M.J.K.); fsysy@mf.seegene.com (J.C.); medsohn@mf.seegene.com (Y.-H.S.); mongguse@naver.com (J.J.L.); 3Department of Laboratory Medicine, Kyung Hee University College of Medicine, Kyung Hee University Hospital, Seoul 02447, Republic of Korea; drkyungsun@gmail.com (K.S.P.); untoyou@hanmail.net (S.Y.C.); 4Department of Infectious Diseases, Kyung Hee University College of Medicine, Kyung Hee University Hospital, Seoul 02447, Republic of Korea; parkkiho@khu.ac.kr

**Keywords:** shotgun metagenomic sequencing, bronchoalveolar fluid (BAL), lower respiratory infection

## Abstract

Conventional diagnostic methods (CDMs) for lower respiratory infections (LRIs) have limitations in detecting causative pathogens. This study evaluates the utility of shotgun metagenomic sequencing (SMS) as a complementary diagnostic tool using bronchoalveolar lavage (BAL) fluid. Sixteen BAL fluid samples from pneumonia patients with positive CDM results—including bacterial/fungal cultures; PCR for *Mycobacterium tuberculosis* or cytomegalovirus; and the BioFire^®^ FilmArray^®^ Pneumonia Panel (BioFire Diagnostics LLC, Salt Lake City, UT, USA)—underwent 10 Gb SMS on the Illumina NovaSeq 6000 platform (Illumina, San Diego, CA, USA). Reads were aligned to the NCBI RefSeq database; with fungal identification further supported by internal transcribed spacer (ITS) analysis. Antibiotic resistance genes (ARGs) were annotated using the Comprehensive Antibiotic Resistance Database. Microbial reads accounted for 0.00002–0.04971% per sample. SMS detected corresponding bacteria in 63% of cases, increasing to 69% when subdominant taxa were included. Fungal reads were low; however, *Candida* species were identified in four samples via ITS. No viral reads were detected. ARGs meeting perfect match criteria were found in two cases. This is the first real-world study comparing SMS with CDMs, including semiquantitative PCR, in BAL fluid for LRI. SMS shows promise as a supplementary diagnostic method, with further research needed to optimize its performance and cost-effectiveness.

## 1. Introduction

Identifying the causative pathogen of infectious pneumonia is crucial for targeted treatment and improved patient outcomes [1]. The detection rate of pathogens in patients with lower respiratory infections (LRIs) ranges from 38–46% [2,3]. Despite the identification of multiple microorganisms using conventional diagnostic methods (CDMs), a single pathogen is typically considered the primary cause of infection [4]. Although polymicrobial infections are reported in 5.7–38.4% of LRI cases, their etiology is rarely confirmed during treatment [5,6,7,8,9].

Bacterial etiology accounts for over 50% of diagnoses, leading to the prioritization of empirical antibiotic use [10]. However, the spectrum of causative microbes is diverse [11]. Culture and polymerase chain reaction (PCR) are standard diagnostic tools supplemented by antigen tests. However, cultures may fail to detect fastidious bacteria and fungi [12,13], while targeted PCR may overlook microbes not included in the assay [14]. Multiplex PCR panels expand the detection range of clinically relevant microbes, enabling comprehensive identification. Nevertheless, these PCR panels have limitations, particularly when pathogens not covered in the panel proliferate and contribute to infection [15]. Antigen tests provide a rapid and cost-effective diagnostic option but are hindered by low sensitivity and false positives [13]. The variability in pathogen detection rates across these diagnostic methods highlights the complexity of achieving accurate LRI diagnoses, emphasizing the need for complementary diagnostic approaches.

In response, shotgun metagenomic sequencing (SMS) has gained attention. This approach involves analyzing all nucleic acids within a sample, enabling the comprehensive identification of potential pathogens. Furthermore, the use of extensive reference databases enhances the potential for syndromic testing, offering universal pathogen detection [16]. Studies utilizing SMS on bronchoalveolar lavage (BAL) specimens have demonstrated a sensitivity of 88–97% and a specificity of 15–81% for accurate pathogen identification [17,18,19,20].

Owing to its anatomical location, BAL fluid inherently possesses low microbial biomass [21], posing challenges for microbial signal detection. Therefore, optimizing methodologies is crucial for achieving reliable SMS results. Despite the development of semiquantitative multiplex PCR, few studies have compared its performance to that of SMS. Furthermore, discussions on the appropriate capacity for SMS of BAL fluid remain limited. The absence of consensus on the criteria for interpreting SMS positivity further complicates the issue [17,18,19,20,22,23,24,25,26].

This study presents the first comparative assessment of SMS and conventional diagnostic tests in clinical practice, including semiquantitative multiplex PCR. The performance of SMS with a 10 Gb output was evaluated, along with efforts to optimize the criteria for SMS interpretation.

## 2. Materials and Methods

### 2.1. Sample Collection and Processing

From March to July 2023, a total of 44 BAL fluid samples with positive results were consecutively collected from the BioFire^®^ FilmArray^®^ Pneumonia Panel (FA-PP; BioFire Diagnostics LLC, Salt Lake City, UT, USA). The FA-PP is a semi-quantitative PCR test representing the latest diagnostic method for pathogen detection. Exclusion criteria for sample selection were as follows: (1) clinical diagnosis of non-infectious pneumonia, (2) contamination with normal flora [27], and (3) final diagnosis of pneumonia caused by RNA viruses. Initially, 12 cases diagnosed with non-infectious diseases were excluded, leaving 32 cases for further analysis (Figure 1).

CDMs were performed using standard clinical protocols for pathogen detection, including bacterial, fungal, and *Mycobacterium tuberculosis*/non-tuberculous mycobacteria (MTB/NTM) cultures. Additional tests included matrix-assisted laser desorption/ionization time-of-flight mass spectrometry system (MADLI-TOF MS; Bruker Daltonics, Billerica, MA, USA), TB/NTM PCR (AdvanSure™ TB/NTM real-time PCR kit on AdvanSure SLAN 96 and E3 system; LG Life Science, Seoul, Korea), Xpert MTB/RIF assay (Cepheid, Sunnyvale, CA, USA), cytomegalovirus (CMV) PCR (nucleic acid extracted using QiaAmp DSP DNA mini kit on QIAcube; Qiagen GmbH, Hilden, Germany, and PCR performed on CFX 96; Bio-Rad, Hercules, CA, USA) and *Pneumocystis jirovecii* PCR (nucleic acid extracted using a laboratory-developed test reagent on MagNa Pure 96; Roche Diagnostics, Mannheim, Germany, and PCR performed on CFX 96; Bio-Rad, Hercules, CA, USA). Antigen detection tests included PLATELIA *Aspergillus* Ag (Bio-Rad, Hercules, CA, USA), targeting galactomannan of *Aspergillus* spp., and Pastorex Crypto Plus (Sanofi-Diagnostics Pasteur, Marnes-La-Coquette, France), detecting cryptococcal capsular polysaccharide glucuronoxylomannan (Appendix A). Antimicrobial susceptibility testing (AST) for cultured isolates was performed using the MicroScan WalkAway 96 plus system (Beckman Coulter, Brea, CA, USA) with NM44, PM28, and MSTRP+1 panels to determine minimum inhibitory concentrations (MICs). The positive reporting criterion for microbial growth was defined as greater than 10^4^ CFU/mL in culture [27]. Species identification was assigned to isolates with MALDI-TOF MS scores ≥ 2.0 [28,29]. FA-PP positivity was defined as exceeding 10^4^ copies/mL. TB/NTM PCR targeted the IS6110 region specific to the MTB complex and the internal transcribed spacer (ITS) region of mycobacteria, using a Ct cutoff of 35. The Xpert MTB/RIF assay detected IS1081 and IS6110 sequences and assessed rifampin resistance based on rpoB target probes. CMV PCR employed a TaqMan probe labeled with VIC and MGBNFQ, with Ct < 40 indicating positive amplification. *P. jirovecii* PCR was interpreted using a Ct threshold of 35.

Based on culture results, 14 samples with a high risk of contamination were additionally excluded, following these criteria:(1)Specimens with a high presence of oropharyngeal normal flora, such as *Streptococcus mitis*, *Streptococcus australis*, *Streptococcus parasanguinis*, *Streptococcus sanguinis*, *Streptococcus clone*, *Streotpcoccus gordonii*, *Streptococcus intermedius*, *Gemella haemolysans*, *Gemella sanguinis*, *Granulicatella adiacens*, *Granulicatella elegans*, *Abiotrophia defective*, and *Rothia dentocariosa*, were excluded. These microorganisms were identified and presumed to be contaminants during the BAL procedure [30,31,32].(2)Specimens with the identification of cutaneous normal flora, such as *Cutibacteriium acnes*, *Cutibacterium granulosum*, *Corynebacterium striatum*, *Staphylococcus epidermidis*, *Staphylococcus hominis*, *Staphylococcus haemolyticus*, *Staphylococcus captis*, *Staphylococcus warneri*, *Staphylococcus saprophyticus*, *Staphylococcus cohnnii*, *Staphylococcus xylosus*, *Staphylococcus simulans*, *Micrococcus luteus*, and *Micrococcus varians*, were excluded. These microorganisms were identified and presumed to pose a risk of contamination during specimen collection and processing [31,33,34,35,36].(3)Commensal microorganisms, including *Staphylococcus aureus*, were identified in the study. Despite their classification as commensals, their potential to cause pneumonia was considered significant [37,38].

These samples included samples containing normal oropharyngeal flora, which may have been introduced during the BAL procedure [32], and normal cutaneous flora, which could have led to contamination during specimen collection and processing [36]. However, commensal microorganisms with a high potential to cause pneumonia, such as *Staphylococcus aureus*, as well as cases in which pathogens such as TB were co-detected, were included in the study [37].

During preliminary nucleic acid extraction experiments conducted for this study, RNA did not meet the minimum quality threshold (RIN ≥ 5), whereas DNA consistently showed DIN ≥ 3, allowing stable library preparation. Therefore, SMS was performed using DNA. Consequently, two FA-PP–positive samples that were clinically diagnosed as infectious pneumonia but yielded only RNA viruses as pathogens were excluded. Ultimately, 16 samples were included in the final analysis.

### 2.2. Extraction of Nucleic Acids and Sequencing

DNA was extracted from 16 BAL fluid samples using the QIAamp DNA Mini Kit (Qiagen, Hilden, Germany). The extracted nucleic acid concentration was measured using the Qubit dsDNA HS Assay Kit on a Qubit 4.0 instrument (Life Technologies, Carlsbad, CA, USA) and the Nanodrop One (Thermo Fisher Scientific, Waltham, MA, USA). DNA integrity was assessed using the Genomic DNA ScreenTape Assay Kit on a TapeStation 4200 System (Agilent Technologies, Inc., Santa Clara, CA, USA).

Libraries were constructed using Illumina DNA Prep (Illumina, San Diego, CA, USA). Quality control was performed using the TapeStation 4200 System with the D1000 ScreenTape Assay Kit (Agilent Technologies, Inc.).

Shotgun sequencing was performed using a paired-end configuration with 150 bp × 2 reads on the NovaSeq 6000 instrument (Illumina, San Diego, CA, USA), utilizing SP Flow Cell reagents and BCL Convert.

### 2.3. SMS Procedures

The sequencing output was set to a capacity of 10 Gb. Following BCL Convert processing, FastQC software (v0.12.0) was used to assess raw data quality by evaluating the proportion of reads with a quality score ≥ Q30 and the GC content. Adapter sequences were largely removed by BCL Convert with high accuracy. Therefore, no additional trimming was performed to avoid excessive data loss, considering both data yield and quality.

Raw reads were mapped to the human reference genome (GRCh38.p14) using URMAP and Samtools, and unmapped reads were extracted and merged using BBMap. Taxonomic assignment of the reads was performed using MMseqs2 against the SILVA_138 database, applying an initial e-value cutoff of 1 × 10^−5^ to identify unique genes across ecosystems. Parameters for taxonomic classification included a minimum alignment length of 90 bp and a minimum identity of 80%, with the best alignment prioritized.

All unmapped reads were assembled using SPAdes with the metaviral option enabled to recover scaffolds corresponding to viral genomes. Reference viral genomes relevant to LRIs were included, such as adenovirus, CMV, Epstein–Barr virus, varicella-zoster virus, and herpes simplex virus [3,11,39,40]. Viral reads were matched to reference viral genomes using MMseqs2 for taxonomic annotation [41].

For further taxonomic classification of shotgun reads, both small subunit (SSU) and internal transcribed spacer (ITS) regions were aligned against the SILVA_138 database using MMseqs2 and analyzed using STAMP [42]. Broader microbial classification was performed using microbial genome databases from the NCBI RefSeq database (release 13 October 2023). Bacterial sequences not classifiable to the genus or species level were labeled as unclassified. Fungal sequences were analyzed using ITS targets, especially in samples with confirmed fungal growth or PCR positivity.

To profile the resistome, antibiotic resistance genes (ARGs) were identified using Resistance Gene Identifier (RGI) software (v6.0.3) and the CARD database (v3.2.8). Identified genes were annotated based on the CARD model. The “perfect” algorithm of RGI, which detects 100% matches to curated reference sequences and known resistance-conferring mutations, was used as the default threshold [42].

Criteria for pathogen detection by SMS reads were established based on prior literature. For bacterial detection, a relative abundance of ≥30% was considered positive [17,18,20,26]. MTB was considered positive with ≥1 mapped read [17,18,19,22,23,24,26], while NTM were considered positive if ranked among the top 10 most abundant bacterial taxa [23,24]. For fungal detection, a species was considered positive when its coverage was at least fivefold higher than that of any other fungal species [22,23,26]. In cases with low coverage, fungal reads were considered positive if alignment length was ≥100 bp and identity ≥ 98%. Viral detection was defined as ≥3 mapped reads [18,24,43].

Microbial reads that met these thresholds were classified as SMS-positive. Subdominant microbes below the threshold were also reviewed. The microbe deemed clinically relevant by the attending physician—based on treatment decisions—was considered the causative agent.

ARG results were compared with AST results from cultured isolates. If the AST confirmed resistance to antibiotics associated with the identified ARGs, the result was deemed consistent. If the isolate showed susceptibility or indeterminate results to any antibiotics in the relevant class, the finding was considered inconsistent. Only “perfect hits”—defined as sequences with 100% identity to reference ARGs—were used as the threshold in this study, based on prior evidence showing 89–95% concordance with culture-based AST; “strict hits” (≥80% identity) showed lower concordance (65%) and were not used [44,45,46].

## 3. Results

### 3.1. Identification of Microbes by SMS

Among the 16 FA-PP-positive samples, the number of reads mapped to the human genome ranged from 34,553,153 to 55,464,668. Microbial reads were detected in all cases, comprising 0.00002–0.04971% of the total reads. Of these, 99.3% were bacterial reads matched to the 16S rRNA gene, 0.3% were eukaryotic reads matched to the 18S rRNA gene, and 0.4% were fungal reads matched to the ITS gene.

SMS detected pathogens above the threshold in 10 of 16 cases (63%), increasing to 11 of 16 cases (69%) when subdominant taxa were included. CDMs identified bacterial pathogens in 11 of 16 cases (69%), including two cases of MTB. Among the 16 cases, bacterial–fungal co-infections were observed in 2 of 16 cases (12.5%), a bacterial–viral co-infection in 1 case (6.25%), and a fungal–viral co-infection in 1 case (6.25%). In the 11 cases in which CDMs identified bacteria as the causative pathogen, SMS detected the corresponding bacteria in nine cases. These included *Pseudomonas aeruginosa* in cases 3, 12, 14, and 15; *Haemophilus influenzae* in cases 10 and 11; *Klebsiella pneumoniae* in case 7; *Acinetobacter baumannii* in case 13; and *Stenotrophomonas maltophilia* in case 16. Expanding the criteria to include subdominant bacterial reads, *K. pneumoniae* was additionally detected in case 8, increasing the detection rate to 10 out of 11 bacterial cases. In cases where fungi were the primary pathogen, SMS detected *Candida tropicalis* reads in case 4 (Table 1).

### 3.2. Metagenomic Results of Antibiotic Resistance

Results meeting the perfect criteria of the RGI were observed in cases 2 and 14.

In case 2, *K. pneumoniae* was evaluated against AST results. The *sul1* gene, associated with sulfonamide resistance, was detected and corresponded with resistance to trimethoprim/sulfamethoxazole in AST. No carbapenem-related ARGs were identified, which was consistent with AST results showing susceptibility to doripenem, ertapenem, imipenem, and meropenem. However, *AAC(6′)-Ia*, associated with aminoglycoside resistance, was detected but was inconsistent with AST results, which showed susceptibility to amikacin and gentamicin (Table 2).

In case 14, *P. aeruginosa* infection was evaluated. *MexI* and *H-NS*, which are associated with fluoroquinolone and tetracycline resistance, were detected and were consistent with AST results showing resistance to levofloxacin and tetracycline. However, several inconsistencies were observed. *OXA-217*, *smeR*, *H-NS*, and *CTX-M-15*, which are associated with resistance to penicillin derivatives and cephalosporins, were detected. However, AST results showed resistance only to ampicillin, piperacillin, and cefotaxime, while susceptibility was observed for cefepime and ceftazidime. For carbapenem resistance, *AXC-1* and *OXA-217* were identified; however, AST results indicated susceptibility to doripenem, imipenem, and meropenem. Additionally, *AAC(6′)-Ia*, *cpxA*, and *smeR*, linked to aminoglycoside resistance, were detected, whereas AST results showed susceptibility to amikacin and an indeterminate result for tobramycin (Table 3).

Regarding antibiotic resistance, no bacteriophages were detected in Case 2, whereas *Inoviridae* species were identified in Case 14.

## 4. Discussion

The proportion of microbial reads averaged 0.00412% of the total reads, ranging from 0.00002–0.04971% per sample. This aligns with the recommended range of 0.00001–0.7% for successful SMS application in clinical settings [47]. Previous studies have established absolute criteria for bacterial positivity in SMS (Table 4), typically defining a relative abundance threshold of ≥30% [17,18,20,26]. However, some cases exhibited SMS-detected reads for subdominant taxa that matched the primary pathogen identified by CDMs. For example, in case 8, *K. pneumoniae* was identified with a relative abundance of 17.78% but was classified as positive using adjusted criteria that considered subdominant taxa. While FA-PP reported *K. pneumoniae* at 10^5^ copies/mL, culture results indicated normal respiratory flora (NRF), which was insufficient to establish it as the primary pathogen. This finding highlights the need for caution when interpreting taxa with relative abundances below 30%, as their classification as pathogens requires additional corroborative evidence. Without clear supporting data, taxa with low relative abundances should not be automatically assumed to represent the primary pathogen. However, their potential clinical relevance warrants careful consideration depending on the clinical context.

Despite applying expanded criteria, SMS did not detect the pathogen in 31% of cases (5 out of 16). This included four cases (involving MTB, *Aspergillus* sp., or CMV) in which no reads were detected, and one case in which FA-PP identified *K. pneumoniae* at 10^5^ copies/mL; however, no corresponding reads were found in SMS.

Although MTB was detected in cases 1 and 2 by CDMs, SMS did not identify any MTB reads. This contrasts with previous studies that report a sensitivity of 44–48% for MTB detection by SMS, comparable to PCR [48,49]. Some studies have reported higher SMS sensitivity than target-specific real-time PCR [50,51]; however, these studies included diverse specimen types, such as lung biopsy tissue, limiting direct comparisons. Several factors may account for the lower sensitivity observed in this study. First, the nucleic acid extraction method used for SMS may have been suboptimal for mycobacteria. A study comparing sputum DNA extraction kits for *Mycobacterium* spp. demonstrated significant differences in 16S rRNA gene cycle threshold (Ct) values depending on the kit used [52]. While no direct comparison exists between the QIAamp DNA Mini Kit used in this study and those used in previous SMS studies, kit-related variations may have influenced the results. The robust, waxy cell wall of *Mycobacterium* spp. makes lysis challenging, often leading to low DNA yield [53]. Additionally, a study on sputum lysis methods for *Mycobacterium* spp. reported differences in Ct value standard deviation based on lysis temperature, further emphasizing the impact of extraction conditions [54]. Second, differences in sample preprocessing may have affected SMS sensitivity. Variations in centrifugal force and duration have been shown to influence *Mycobacterium* sedimentation, affecting recovery rates and smear sensitivity [55]. Notably, our study applied a higher relative centrifugal force than previous studies that reported greater SMS sensitivity for MTB. Excessive centrifugal force can generate heat, potentially leading to bacterial injury and reduced DNA recovery [49,50,55]. The absence of phenol treatment, which has been suggested to enhance DNA purity, may have further contributed to reduced detection [53,56]. Finally, the patients’ treatment history may have influenced SMS sensitivity. Previous studies have reported a significant decrease in MTB detection sensitivity from 76% in pretreatment samples to 31% in post-treatment samples [49]. Both cases 1 and 2 involved patients with a history of anti-tuberculosis treatment, and neither case exhibited rifampin resistance. In case 2, *M. tuberculosis* was detected by the Xpert MTB/RIF assay and cultured; however, *Corynebacterium striatum* was the predominant isolate, suggesting that competition with other organisms and prior treatment may have reduced SMS sensitivity. Enhancing MTB detection may require developing targeted panels optimized for *Mycobacterium* specimen processing.

In case 6, in which *Aspergillus* sp. was detected by CDMs, SMS did not yield any reads, likely due to the incomplete or variable nature of ITS reference sequences, which complicates fungal identification via SMS [57]. In this study, fungal species identified via CDMs were not detected as positive reads using SMS. The ITS1 sequence is recognized for its reliability in identifying *Candida* spp., *Pneumocystis* spp., and *Aspergillus* spp. within the *Ascomycota* division, offering accurate species- and genus-level classification [58]. Although fungal pneumonia is increasingly prevalent among immunocompromised individuals, all cases in this study were HIV-negative [13]. Notably, *P. jirovecii* pneumonia detection has been increasing among HIV-uninfected immunocompromised individuals, highlighting its clinical relevance [59]. However, SMS did not detect *P. jirovecii* reads in this study, even with the complete ITS1 sequence. Previous studies have reported significantly lower quantities of *Pneumocystis* spp. in BAL fluid from HIV-negative patients than from HIV-positive patients, which may have contributed to the challenges in SMS-based detection [60].

For cases 5, 6, and 16, CMV was considered the primary pathogen based on clinical assessment, with PCR results ranging from 9665 to 200,175 copies/mL. In case 4, CMV was detected at 1,343,600 copies/mL; however, it was not targeted for treatment owing to the patient’s immunological status. These findings highlight the limitations of relying solely on CDMs for clinical decision-making, as CMV PCR results require integration with a broader clinical context. Notably, SMS did not detect viral reads in any case, precluding the identification of CMV or other viruses. Despite DNA extraction yields exceeding 3–8 times the 4 nM threshold for library preparation, the proportion of valid viral sequences remained low (~0.05%), indicating potential biases [61]. The small size of viral genomes further complicates SMS-based detection [62]. Although the DNA integrity number exceeded the manufacturer’s recommended threshold of 3, no standardized criteria exist for library construction. Additionally, while the Q30 score surpassed the 85% threshold, it did not reach 90%, which may have affected viral read recovery [63]. Moreover, extended storage and delayed processing for SMS, compared to immediate testing for FA-PP and CMV PCR, may have contributed to viral degradation and reduced detection rates [64]. Bacteriophages, which constitute the majority of viral particles in both environmental and human-associated microbiomes, account for over 90% of the human virome [65,66,67]. Incorporating bacteriophage detection strategies into viral metagenomic analyses may enhance SMS-based viral identification.

All bacteria detected as positive by SMS in this study were Gram-negative. Gram-negative bacilli are the most common causative agents of LRIs in elderly patients [4], and the 10 Gb SMS platform demonstrated strong performance in detecting these pathogens. In Case 2, where *Corynebacterium striatum* was the predominant isolate by culture, SMS revealed *Serratia* spp. as the second most abundant taxon. Among these, reads corresponding to *S. marcescens* (26 reads, 6.25%)—a known cause of LRI in hospitalized patients—were detected, along with *S. rubidaea* (4 reads, 1%). *S. rubidaea* is a rare but emerging pathogen capable of causing LRIs, as reported in several case studies [68]. Due to its overlapping biochemical characteristics with lactose-fermenting *Enterobacterales*, *S. rubidaea* is difficult to identify using CDMs [68,69]. This case illustrates how SMS can aid in the detection of potentially pathogenic bacteria that are difficult to isolate or identify via culture due to sample-related or organism-specific limitations and the 10 Gb SMS demonstrated strong performance in detecting these pathogens.

Unlike previous studies, this study incorporated FA-PP into CDMs, potentially enhancing the pathogen detection rate. FA-PP has been shown to identify target pathogens even in samples reported as “no growth” or “normal flora” in culture [70]. However, no clear correlation was observed between FA-PP semi-quantitative values and SMS read numbers. For FA-PP values of 10^5^ copies/mL, SMS reads ranged from 5–52; for 10^6^ copies/mL, reads ranged from 2–310; and for values exceeding 10^7^ copies/mL, reads ranged from 347–23,869. A culture result of 10^4^ CFU/mL corresponded to 10^5^ copies/mL in FA-PP, 3000–6000 CFU/mL corresponded to 10^6^ copies/mL, and ≥10^5^ CFU/mL aligned with 10^6^–10^7^ copies/mL in FA-PP.

This study partially investigated genotypic AST using SMS in BAL fluid; however, the results did not consistently correlate with phenotypic AST of cultured isolates. Using the CARD database, SMS-detected ARGs were compared with phenotypic AST results from CDM-identified pathogens [44]. In previous studies, the use of the perfect threshold showed a discordance rate of only 8%, significantly lower than the 35% observed with the *strict* threshold, suggesting that the perfect criterion reduces false positives and improves clinical reliability [45,46]. The strict threshold has been criticized for potentially identifying non-functional genes unrelated to resistance phenotypes, which may mislead treatment decisions. Conversely, the perfect threshold is considered more suitable for providing actionable insights for targeted therapy, as it minimizes overprediction of resistome diversity and accounts better for clinical relevance [45,46]

Despite applying only the perfect algorithm in this study, ARGs could not be conclusively linked to the cultured pathogens. In case 14, *Inoviridae* sp. was detected, suggesting that bacteriophage-mediated mechanisms may have influenced resistance expression or contributed to indeterminate AST results, despite the presence of ARGs. Although *AXC-1* and *OXA-217* were identified, phenotypic carbapenem resistance was not observed. This may be attributed to *Inoviridae* sp. enhancing *P. aeruginosa* biofilm formation and stability, thereby reducing antibiotic penetration [71,72]. Additionally, the indeterminate result for aminoglycosides may be partially explained by reduced outer membrane permeability associated with *cpxA* regulation influenced by *Inoviridae* sp. [73]. The high severity of illness in this case, with a pneumonia severity index (PSI) of 164 (grade 5), also raises the possibility that bacterial virulence was enhanced by bacteriophage interaction [74].

A key limitation of SMS is the inability to trace specific ARGs to their bacterial hosts [75], which may lead to an overestimation of resistance, especially when resistance is confined to a single antibiotic within a class. Although ARGs are detectable at the DNA level, their functional expression at the RNA or protein level remains unknown, limiting insights into actual resistance mechanisms. A meta-analysis of genotypic AST using metagenomic sequencing reported a categorical agreement of 88%; however, very major errors (VME) (24%, 95% CI: 8–40%) and major errors (ME) (5%, 95% CI: 0–12%) exceeded the US Food and Drug Administration (FDA)-recommended thresholds [75]. While a machine learning-based genotypic AST model met FDA requirements (ME ≤ 3%, VME ≤ 1.5%) with high performance [76], clinical specimens pose additional challenges. The presence of host nucleic acids, multiple pathogens, and plasmid-mediated ARGs complicates species attribution [76,77,78]. Additionally, database limitations and low area under the curve (AUC) values in some models highlight the need for further research to refine genotypic AST approaches [76].

However, the direct application of SMS-detected ARGs in clinical practice remains premature. While SMS cannot conclusively determine the direct involvement of ARGs in resistance phenotypes, it enables the detection of microbes at very low concentrations and provides a comprehensive overview of the resistome within a sample [70,75]. The administration of antibiotics to patients colonized with multidrug-resistant organisms increases colonization density and expands the resistance gene pool, thereby increasing the risk of infection [76,79]. The maintenance of resistance-conferring mutations in bacterial pathogens is entirely dependent on their effect on fitness and virulence [80]. Given the association between the resistome and microbial diversity, antimicrobial stewardship is increasingly emphasized. Overcoming current limitations may allow SMS-based ARG identification to contribute to these efforts [76].

This study demonstrates the feasibility of achieving sufficient sequencing coverage for SMS with a 10 Gb output. Previous studies have not provided detailed sequencing configurations for BAL fluid. While highly complex samples such as stool require 1–10 Gb (with >7 Gb recommended), lower-complexity samples such as anterior nares may be adequately analyzed with <1 Gb [81,82]. To assess whether 10 Gb was sufficient for BAL fluid analysis, we conducted experiments based on existing recommendations [47]. Increasing sequencing output generally enhances pathogen detection; however, in untargeted approaches without host genome depletion, a proportional increase in host-derived sequences may limit improvements in sensitivity [83]. Using a 10 Gb configuration, DNA-based SMS detected primary pathogens in 63% of cases based on applied thresholds, increasing to 69% when subdominant taxa were included. In SMS, the presence of host nucleic acid affects sensitivity, and appropriate depletion can increase the relative abundance of microbial signals [84]. However, unintentional removal of microbial content along with host DNA and the variability of results depending on sample type remains controversial [85]. In this study, SMS was conducted without host genome depletion. Further research is required to validate the effective host signal removal in BAL samples.

Inconsistencies between SMS and clinical diagnoses are well-documented, with comparative studies reporting low agreement levels (κ = 0.035–0.347) between SMS and CDM positivity [22,23,26]. In this study, cases with negative SMS results despite positive CDM findings underscore the limitations of SMS as a molecular diagnostic tool [86]. Challenges include differentiating viable pathogens from residual nucleic acids in resolved or treated infections and distinguishing asymptomatic colonization from true infection [14]. Additionally, variations in bacterial read distributions complicate the application of uniform positivity thresholds. Future studies should refine SMS interpretation criteria by integrating clinical features to improve their correlation with clinical outcomes. Comparative analyses of sequencing output capacities for BAL fluid samples are required to determine the optimal conditions for pathogen detection. Further research should also explore methodologies that expand SMS applications beyond the bacteriome to include the mycobiome and virome, thereby enhancing its diagnostic potential.

## 5. Conclusions

This study presents a comparison between SMS and CDMs, elucidating the performance and limitations of SMS with a 10 Gb output for LRI diagnosis. While SMS cannot replace PCR, it can complement culture methods, serving a mutually supportive role. To establish SMS as a valuable supplementary tool in LRI diagnostics, further research is needed to explore its variable applications for higher sensitivity and cost-effectiveness.

Abbreviations: CDM, conventional diagnostic method; BAL, bronchoalveolar lavage; MALDI-TOF MS, matrix-assisted laser desorption/ionization time-of-flight mass spectrometry; FA-PP, FilmArray^®^ Pneumonia Panel; MTB, *Mycobacterium tuberculosis*; NTM, non-tuberculous mycobacteria; MTB, *Mycobacterium tuberculosis*; CMV, cytomegalovirus.

## Figures and Tables

**Figure 1 microorganisms-13-01338-f001:**
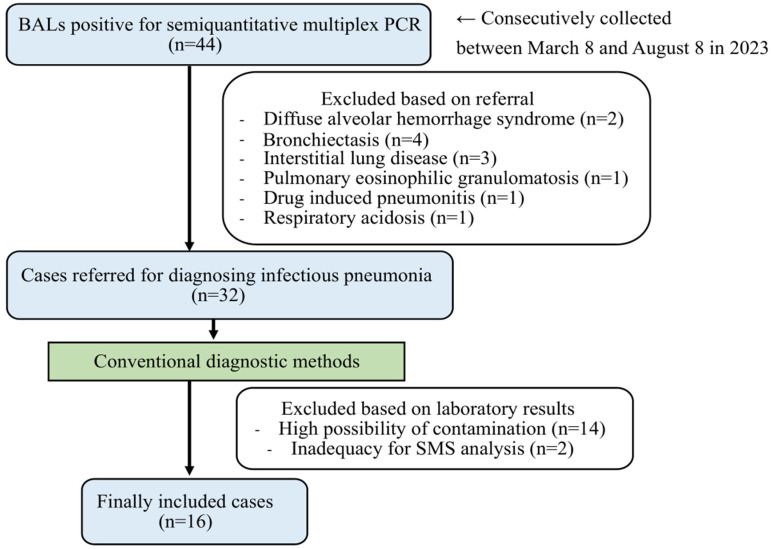
Flow diagram of case inclusion and exclusion. BAL: bronchoalveolar lavage. SMS: shotgun metagenomic sequencing.

**Table 1 microorganisms-13-01338-t001:** Comparison of microbial results from CDMs and SMS.

No	Results of CDM			Results of SMS *	
	Culture (CFU/mL)	Filmarray Pneumonia Panel PCR (Copies/mL)	Singleplex Tests †	Taxon Above Threshold ‡	Subdominant Taxon
1	*Escherichia coli* (60,000)	*Escherichia coli* (10^6^) *Klebsiella pneumoniae *(10^5^)	MTB complex	None	*Streptococcus salivarius* (111, 25.58%)
2	*Klebsiella pneumoniae* (30,000) *Corynebacterium striatum* (≥100,000) MTB complex	*Klebsiella aerogenes *(10^6^) *Staphylococcus aureus *(10^6^)	MTB complex CMV (128,345) *Pneumocystis jirovecii*	None	*Corynebacterium striatum* (110, 26.44%)
3	*Candida albicans* (50,000)	*Pseudomonas aeruginosa* (10^6^) *Escherichia coli* (10^4^) Rhinovirus/Enterovirus	CMV (4395)	*Pseudomonas aeruginosa* (35, 87.5%)	*Lactobacillus fermentum* (2, 5%)
4	*Candida tropicalis* (10,000)	*Enterobacter cloacae* complex (10^4^)	CMV (1,343,600)	*Candida tropicalis* (5)	*Corynebacterium striatum* (1, 100%)
5	NRF	Adenovirus	CMV (9665) *Pneumocystis jirovecii*	None	*Staphylococcus kloosii* (2, 11.11%)
6	NRF	Parainfluenza virus	CMV (200,175) *Aspergillus*(7.68)	None	*Geobacillus stearothermophilus* (1, 25%)
7	*Klebsiella pneumoniae* (10,000)	*Klebsiella pneumoniae* (10^5^)	None	*Klebsiella pneumoniae* (13, 30.23%)	*Serratia marcescens* (3, 6.98%)
8	NRF	*Klebsiella pneumoniae* (10^5^)	CMV (1015)	None	*Klebsiella pneumoniae* (8, 17.78%)
9	NRF	*Klebsiella pneumoniae* (10^5^)	None	None	None
10	NRF	*Haemophilus influenzae* (10^6^) Metapneumovirus	None	*Haemophilus influenzae* (90, 59.60%)	*Haemophilus parasuis* (2, 1.33%)
11	NRF	*Haemophilus influenzae* (10^5^) Influenza A	CMV (<257)	*Haemophilus influenzae* (5, 31.25%)	*Pasteurella multocida* (2, 12.5%)
12	*Pseudomonas aeruginosa* (≥100,000)	*Pseudomonas aeruginosa* (10^6^)	CMV (<257) *Pneumocystis jirovecii* *Aspergillus* (1.08)	*Pseudomonas aeruginosa* (59, 75.64%)	*Pseudomonas* sp. (6, 7.69%)
13	*Acinetobacter baumannii* (≥100,000)	*Acinetobacter calcoaceticus-* *baumannii* complex (10^6^)	*Aspergillus* (2.27)	*Acinetobacter baumannii* (141, 66.82%)	*Acinetobacter nosocomialis* (11, 5.21%) *Candida albicans* (6)
14	*Pseudomonas aeruginosa* (≥100,000)	*Acinetobacter calcoaceticus-* *baumannii* complex (≥10^7^) *Pseudomonas aeruginosa* (≥10^7^) *Escherichia coli * (10^6^) *Serratia marcescens* (10^6^) *Klebsiella pneumoniae* 10^5^)	CMV (<257)	*Pseudomonas aeruginosa* (23,869, 86.63%)	*Acinetobacter baumannii* (347, 1.26%)
15	*Pseudomonas aeruginosa* (≥100,000)	*Pseudomonas aeruginosa* (10^6^)	CMV (2065) *Aspergillus* (1.90)	*Pseudomonas aeruginosa* (24, 75%)	*Pseudomonas* sp. (2, 6.25%)
16	*Stenotrophomonas maltophilia* (≥100,000)	Parainfluenza virus	CMV (10,780)	*Stenotrophomonas maltophilia* (147, 44.41%)	*Stenotrophomonas pavanii* (26, 7.85%)

Detected microorganisms that were treatment targets are underlined. For pathogens detected by both culture and the PCR, only the culture results are underlined. * For SMS results, the read numbers and their fractions among 16S reads were recorded in parentheses for bacteria, while only the read numbers were recorded in parentheses for fungi. † Singleplex tests included PCR for MTB, CMV, and *Pneumocystis jirovecii* as well as an antigen test for *Aspergillus*. Quantitative values for CMV PCR were recorded in parentheses as copies/mL, and antigen test results for *Aspergillus* were also noted in parentheses. ‡ For SMS interpretation, a threshold of 30% relative abundance, based on the fraction of 16S reads, was used as per previous studies. Abbreviations: no., number; CDM, conventional diagnostic method; SMS, shotgun metagenomic sequencing; MTB; Mycobacterium tuberculosis; CMV, cytomegalovirus; NRF, normal respiratory flora.

**Table 2 microorganisms-13-01338-t002:** Antibiotic resistance genes detected in case 2 with Klebsiella pneumoniae.

Antibiotic Group	Resistance Gene	Gene Detection Status	Antibiotic Susceptibility	Consistency *
Carbapenem	None	Not detected	Doripenem (S), Ertapenem (S), Imipenem (S), Meropenem (S)	Consistent
Aminoglycoside	*AAC(6′)-Ia*	Detected	Amikacin (S), Gentamicin (S)	Inconsistent
Sulfonamide	*Sul1*	Detected	Trimethoprim/Sulfamethoxazole (R)	Consistent

* The antibiotic resistance gene aligned with the resistance profile of the cultured isolates. Abbreviations: S, susceptible; R, resistant.

**Table 3 microorganisms-13-01338-t003:** Antibiotic resistance genes detected in case 14 with Pseudomonas aeruginosa.

Antibiotic Group	Resistance Gene	Gene Detection Status	Antibiotic Susceptibility	Consistency *
Carbapenem	*AXC-1*, *OXA-217*	Detected	Doripenem (S), Imipenem (S), Meropenem (S)	Inconsistent
Aminoglycoside	*AAC(6′)-Ia*, *cpxA*, *smeR*	Detected	Amikacin (S), Tobramycin (I)	Inconsistent
Fluoroquinolone and Tetracycline	*MexI*, *H-NS*	Detected	Levofloxacin (R), Tetracycline (R)	Consistent
Penicillin derivatives and Cephalosporins	*OXA-217*, *smeR*, *H-NS*, *CTX-M-15*	Detected	Ampicillin (R), Piperacillin (R), Cefotaxime (R), Cefepime (S), Ceftazidime (S)	Inconsistent

* The antibiotic resistance gene aligned with the resistance profile of the cultured isolates. Abbreviations: S, susceptible; R, resistant; I, indeterminate.

**Table 4 microorganisms-13-01338-t004:** Thresholds for identifying clinically significant microbe using SMS.

Disease	Sequencing	Principles of Setting Thresholds	Thresholds for Identifying Pathogen by SMS				Study
			Bacteria	Mycobacteria	Fungi	Viruses	
Suspected pulmonary infection	BGISEQ (BGI, Shenzhen, China)		≥50 unique reads from a single species	≥1 unique read from MTBC	≥50 unique reads from a single species	≥50 unique reads from a single species	[19]
Suspected pulmonary infection	BGISEQ-100 (BGI, China)	Reads number ≥ 50 and pathogen detected by traditional method	>30% relative abundance at the genus level	≥1 unique read from MTB		SMRN ≥ 3	[18]
Suspected pulmonary infection	MiniSeq (Illumina, USA)		>30% relative abundance at the genus level, or histopathological examination and/or culture positive with ≥50 unique reads of a single species	≥1 unique read from MTB	>30% relative abundance at the genus level, or histopathological examination and/or culture positive with ≥50 unique reads of a single species	>30% relative abundance at the genus level	[17]
Suspected pneumonia (immunocompromised)	Agilent 2100 Bioanalyzer (Thermo Fisher Scientific, USA)	Regardless of coverage rate, oral commensals were not defined as CSMs unless they were deemed to be significant by the physicians or proven otherwise	Coverage rate ≥ 10 times any other microbes	≥1 unique read from MTB Mapping read number in the top 10 in the bacteria list of NTM	Coverage rate ≥ 5 times any other fungus	Coverage rate ≥ 10 times any other microbes	[23]
Pneumonia	Bioelectron Seq 4000 (CapitalBio Corporation, Beijing, China)	Clinically pathologic microorganism is defined; -Definite: SMS result is consistent with results from CDMs (culture, nucleic acid-based testing, and pathological examination) Probable: SMS pathogen is likely the cause of pneumonia according to clinical, radiologic, or laboratory findings, but the SMS result was consistent with CDMs.	Coverage rate ≥ 10 times any other microbes	≥1 unique read from MTB	Coverage rate ≥ 5 times any other fungus	Coverage rate of species level ≥ 10 times any other microbes	[22]
Severe pneumonia (immunocompromised)	KAPA Library Quantification 75-cycle sequencing kit (Illumina, USA)	>30% Relative abundance at the genus level, Reads number ≥ 50 from a single species and pathogen detected by culture					[20]
Community-acquired pneumonia	Nextseq 550Dx (Illumina, USA)	RPM counts ≥ 5 times the values of the NEC				Coverage of ≥3 non-overlapping regions on the genome	[25]
Community-acquired pneumonia	NextSeq CN500 (Illumina, USA)		Reads number ≥ 50 or pathogen detected by culture	≥1 unique read from MTB Mapping read number in the top 10 in the bacteria list of NTM	≥3 reads mapped to pathogen species, or supported by clinical culture	≥3 reads mapped to pathogen species, or supported by clinical culture	[24]
Severe community-acquired pneumonia (immunocompromised)	N/A	Reads number ≥ 50 from a single species and pathogen detected by culture	>30% Relative abundance at the genus level, Coverage rate ≥ 10 times any other bacteria	≥1 unique read from MTB	>30% relative abundance at the genus level; Coverage rate ≥ 5 times any other fungus		[26]

Abbreviations: SMS, shotgun metagenomic sequencing; RPM, read per million; NEC, negative extraction control; MTBC, Mycobacterium tuberculosis complex; MTB, Mycobacterium tuberculosis; SMRN, stringently mapped read number; CSMs, clinically significant microbes; NTM, non-tuberculous mycobacteria; CDM, conventional diagnostic method; N/A, not accessible.

## Data Availability

The raw sequencing data generated in this study have been deposited in the NCBI Sequence Read Archive (SRA) under the accession number PRJNA1036216.

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
