# Peer review of "Shotgun Metagenomic Sequencing Analysis as a Diagnostic Strategy for Patients with Lower Respiratory Tract Infections"

_microorganisms, 2025, doi:10.3390/microorganisms13061338_

Round 1
Reviewer 1 Report
Comments and Suggestions for Authors
The manuscript “Shotgun metagenomic sequencing analysis as a diagnostic strategy for patients with lower respiratory tract infections” by Cho et al. compares the effectiveness of shotgun metagenomic sequencing (SMS) with conventional diagnostic methods (CDMs) for identifying causative pathogens in lower respiratory infections (LRIs). The study used bronchoalveolar lavage (BAL) fluid samples from pneumonia patients with positive CDM results. SMS detected corresponding bacteria in 63% of cases, increasing to 69% when subdominant taxa were included. Fungal reads were low, and no viral reads were detected. Antibiotic resistance genes were found in two cases. The authors conclude that SMS shows promise as a supplementary diagnostic method for LRI, but further research is needed to optimize its performance and cost-effectiveness.
While the article presented holds promise, aspects of it would benefit from additional refinement to enhance their robustness and rigor. Specific areas for improvement are outlined below:
Materials and Methods:
Lines 89-90: “Additional tests included matrix-assisted laser desorption/ionization…”. Because this is a method comparison study, the authors could include information on the sensitivity and specificity of each of these tests used as a supplementary table.
Lines 103-104: “Based on culture results, 14 samples with a high risk of contamination were additionally excluded”. How were these contamination exclusion criteria measured and quantified? Was a minimum threshold considered?
Lines 110-111: “Given that this study utilized DNA-based SMS and did not target RNA, two additional samples in which only RNA viruses were identified via CDMs were excluded”. Please indicate which tests allowed this identification.
Lines 124-126: “Shotgun sequencing was performed using a paired-end configuration with 150 bp × 2 reads on the NovaSeq 6000 instrument (Illumina, San Diego, CA, USA), utilizing SP Flow Cell reagents”. This sentence could be moved to the "Extraction of nucleic acids and library preparation" section and renamed "Extraction of nucleic acids and sequencing"
Line 129: “Adapter sequences were removed through trimming”. What program was used to perform adapter trimming and remove bases from reads with a quality lower than 30?
Lines 139-140: “Databases were downloaded from the National Center for Biotechnology Information (NCBI) RefSeq (version 2023.10.13) for classification”. Could you please elaborate on the classification method employed? Was blastn or another alignment tool utilized in this process?
Line 147: “The criteria for pathogen detection in SMS were applied based on existing literature”. This part of the methodology is confusing. Was taxonomic identification performed at the contig or read level? And in the case of reads, which program was used? Regardless of the method used, at this point I have some doubts about the reliability of the detection method used in the metagenomic data. In most cases, taxonomic identification of shotgun data is performed at the read level using recommended programs for this purpose, such as Kraken2, Kaiju, or MetaPhlAn, which use different approaches and databases. Why didn't the authors use those programs and perform a comparative analysis, along with the method they used in this article?
Line 160: “The identified ARGs were determined using the "perfect" algorithm of the RGI…”. Why wasn't the strict option used? I think using the perfect method would be missing out on relevant information. It would be interesting if the authors also considered using the RGI bwt that allows mapping reads to reference genes.
Results:
Line 168: “Identification of microbes by SMS”. The authors could indicate how many reads per sample mapped to the human genome.
Line 196: “Metagenomic results of antibiotic resistance”. The authors report that in some cases there is no correlation between the RGI and AST results. Could this be a consequence of using the perfect RGI parameter? The authors could also consider using the RGI option that allows direct use of reads instead of contigs.
It would also be advisable for the authors to identify virulence genes in their data (either at the contig or read level) and to correlate them with the symptoms observed in each patient from whom the samples were obtained.
Comments on the Quality of English LanguageThe manuscript requires revision by a native language specialist to address grammatical errors and improve paragraph structure.
Author Response
We sincerely thank Reviewer 1 for your thoughtful and constructive comments. We appreciate your recognition of the potential in our work, as well as your detailed suggestions for improving its scientific rigor and clarity. In response, we have carefully reviewed each of your recommendations and revised the manuscript accordingly to enhance the robustness and methodological transparency of our study.
All suggested revisions have been addressed point-by-point in the following responses, and the relevant sections of the manuscript have been updated. We are grateful for the opportunity to refine our work through your insightful feedback and hope that the revised version meets your expectations.
Materials and Methods:
Comments 1: Lines 89-90: “Additional tests included matrix-assisted laser desorption/ionization…”. Because this is a method comparison study, the authors could include information on the sensitivity and specificity of each of these tests used as a supplementary table.
Response 1: We appreciate the reviewer’s valuable suggestion. In response, we have included a supplementary table summarizing the sensitivity and specificity of each method to better support the comparative nature of this study. Please refer to the Supplementary Materials section (lines 457 and onward).
Comments 2: Lines 103-104: “Based on culture results, 14 samples with a high risk of contamination were additionally excluded”. How were these contamination exclusion criteria measured and quantified? Was a minimum threshold considered?
Response 2: Thank you for this important point. We have clarified the contamination exclusion criteria in the revised manuscript. The decision was based on the likelihood that the detected microbes represented commensal flora with low pathogenic potential. Given the nature of BAL specimens and the sampling process, we excluded microbial species considered highly likely to reflect contamination rather than true infection. This has been elaborated in lines 113–129.
Comments 3: Lines 110-111: “Given that this study utilized DNA-based SMS and did not target RNA, two additional samples in which only RNA viruses were identified via CDMs were excluded”. Please indicate which tests allowed this identification.
Response 3: We thank the reviewer for pointing out the lack of clarity. We have now specified that this decision was based on findings from prior pilot studies assessing nucleic acid integrity for library preparation. Additionally, we clarified that RNA viruses were identified using the FA-PP test and explicitly excluded two cases where only RNA viruses were detected. This has been addressed in lines 136–140.
Comments 4: Lines 124-126: “Shotgun sequencing was performed using a paired-end configuration with 150 bp × 2 reads on the NovaSeq 6000 instrument (Illumina, San Diego, CA, USA), utilizing SP Flow Cell reagents”. This sentence could be moved to the "Extraction of nucleic acids and library preparation" section and renamed "Extraction of nucleic acids and sequencing"
Response 4: We appreciate the reviewer’s editorial suggestion. As recommended, we have reorganized this sentence under the section now titled “Extraction of nucleic acids and sequencing” to enhance logical flow. Please see lines 142 and 152–154.
Comments 5: Line 129: “Adapter sequences were removed through trimming”. What program was used to perform adapter trimming and remove bases from reads with a quality lower than 30?
Response 5: Thank you for highlighting this point. We have clarified that adapter trimming and base quality filtering were performed during the BCL Convert process. Upon further review, no additional trimming was performed to avoid unnecessary data loss. This is now specified in lines 154 and 156–160.
Comments 6: Lines 139-140: “Databases were downloaded from the National Center for Biotechnology Information (NCBI) RefSeq (version 2023.10.13) for classification”. Could you please elaborate on the classification method employed? Was blastn or another alignment tool utilized in this process?
Response 6: We thank the reviewer for the opportunity to clarify this important methodological aspect. We have now specified the tools used for taxonomic classification and elaborated on the steps of metagenomic data analysis. Details are provided in lines 162–163 and 170–178.
Comments 7: Line 147: “The criteria for pathogen detection in SMS were applied based on existing literature”. This part of the methodology is confusing. Was taxonomic identification performed at the contig or read level? And in the case of reads, which program was used? Regardless of the method used, at this point I have some doubts about the reliability of the detection method used in the metagenomic data. In most cases, taxonomic identification of shotgun data is performed at the read level using recommended programs for this purpose, such as Kraken2, Kaiju, or MetaPhlAn, which use different approaches and databases. Why didn't the authors use those programs and perform a comparative analysis, along with the method they used in this article?
Response 7: Thank you for pointing out the need for clarification. We have revised the sentence to more clearly explain that we applied pathogen identification criteria to microbial reads already classified through prior bioinformatics steps. The clarification is in lines 184–185. Details on taxonomic identification tools, as addressed in Response 6, are also available in lines 162–163 and 170–178. We appreciate the reviewer’s attention to the analytical rigor of our metagenomic pipeline.
Comments 8: Line 160: “The identified ARGs were determined using the "perfect" algorithm of the RGI…”. Why wasn't the strict option used? I think using the perfect method would be missing out on relevant information. It would be interesting if the authors also considered using the RGI bwt that allows mapping reads to reference genes.
Response 8: We appreciate this insightful comment. In our study, we prioritized comparison with culture-based AST results and thus opted for the “perfect” algorithm to enhance clinical relevance and specificity. While we recognize the value of including both perfect and strict parameters, we aimed to focus on actionable resistance markers consistent with clinical observations. This rationale is now explained in lines 199–203. Additionally, we further discussed this issue in the Discussion section (lines 377–386) in light of the reviewer’s important suggestion.
Results:
Comments 9: Line 168: “Identification of microbes by SMS”. The authors could indicate how many reads per sample mapped to the human genome.
Response 9: Thank you for the suggestion. We have now included information on the number of human genome–mapped reads per sample to provide additional context for the metagenomic findings. Please refer to lines 206–207.
Comments 10: Line 196: “Metagenomic results of antibiotic resistance”. The authors report that in some cases there is no correlation between the RGI and AST results. Could this be a consequence of using the perfect RGI parameter? The authors could also consider using the RGI option that allows direct use of reads instead of contigs.
It would also be advisable for the authors to identify virulence genes in their data (either at the contig or read level) and to correlate them with the symptoms observed in each patient from whom the samples were obtained.
Response 10: We fully agree that including both “perfect” and “strict” RGI parameters could offer a more comprehensive analysis. However, even with both criteria, only two cases (Cases 2 and 14) yielded identifiable ARGs, and the majority of strict hits corresponded to antibiotics not included in our AST panel. Given our focus on correlating metagenomic data with clinically relevant culture results, we proceeded with the “perfect” threshold alone. In response to the reviewer’s excellent suggestion, we have incorporated phage analysis in Case 14, where Inoviridae sp. was detected and may have influenced resistance expression. This result is noted in lines 260–261, and further discussed in lines 386–395. The implications of algorithm choice are discussed in lines 377–386.
Response to Comment on Language and Grammar:
We sincerely thank Reviewer 1 for pointing out the need for language refinement. The original manuscript was reviewed by a native English editing service prior to submission. However, we acknowledge that some grammatical inconsistencies may have been introduced during subsequent revisions. In response to your valuable suggestion, we have carefully re-reviewed the entire manuscript and made necessary corrections to ensure that the language, grammar, and paragraph structure meet the standards expected of a scientific publication. We are truly grateful for your thorough and thoughtful review, which has helped us improve the clarity and overall quality of the manuscript.
Once again, we are sincerely grateful for Reviewer 1’s careful review and constructive recommendations, which have significantly improved the quality and clarity of our manuscript. We hope that the revisions and clarifications now meet your expectations.

Reviewer 2 Report
Comments and Suggestions for Authors
Interesting manuscript with novel information on SMS application in clinical diagnostics. The methods section however lacks details on the culture, singleplex PCR and AST protocols utilized in the study. Apart from the targeted microbes for the SMS (FA-PP/CDM/culture confirmed) based using the threshhold and subdominant taxa, were there other unique microbes and ARGs detected by SMS? This could added to the results section and further discussed.
Author Response
Reply to the Review Report to Reviewer 2
We sincerely thank Reviewer 2 for your encouraging feedback and insightful comments. Your remarks have greatly helped to improve the clarity and completeness of our manuscript. We have addressed each of your comments below.
Materials and Methods:
Comments 1: The methods section however lacks details on the culture, singleplex PCR and AST protocols utilized in the study.
Response 1: We thank the reviewer for pointing out this important omission. In response, we have revised and expanded the Methods section to include more detailed descriptions of the conventional diagnostic methods (CDMs). Specifically, we clarified the criteria for applying MALDI-TOF MS to cultured isolates (lines 104–105), described the multiplex and singleplex PCR protocols (lines 105–111), and included information on the antigen detection assays (lines 98–100). Details regarding the AST methodology were also added (lines 101–103). We believe these additions enhance the methodological transparency of our study.
Discussion:
Comments 2: Apart from the targeted microbes for the SMS (FA-PP/CDM/culture confirmed) based using the threshold and subdominant taxa, were there other unique microbes and ARGs detected by SMS? This could added to the results section and further discussed.
Response 2: We appreciate the reviewer’s thoughtful suggestion. In response, we have added discussion on additional microbial findings beyond the primary threshold and subdominant taxa. In particular, we highlighted a case where Corynebacterium striatum was not only dominant in SMS reads but also grew prominently in culture. We discuss the potential clinical relevance and the added value of SMS in identifying such organisms that may otherwise be underestimated. This discussion has been included in lines 355–364.
Once again, we sincerely thank Reviewer 2 for the valuable feedback and for recognizing the novelty and clinical relevance of our study. We hope the revised manuscript meets your expectations.

Round 2
Reviewer 1 Report
Comments and Suggestions for Authors
I have reviewed the resubmission of the manuscript entitled "Shotgun metagenomic sequencing analysis as a diagnostic strategy for patients with lower respiratory tract infections". The authors answered in a satisfactory way the points that I have addressed in the first review. Thus this new version of the manuscript can be accepted for publication.